# Remote Sensing Imaging as a Tool to Support Mulberry Cultivation for Silk Production

Domenico Giora [1,*], Alberto Assirelli [2], Silvia Cappellozza [3], Luigi Sartori [1], Alessio Saviane [3], Francesco Marinello [1] and José A. Martínez-Casasnovas [4,5]

1  Department of Land, Environment, Agriculture and Forestry, University of Padova, Agripolis, Legnaro, 35020 Padova, Italy
2  Council for Agricultural Research and Economics, Research Centre for Engineering and Agro-Food Processing, Monterotondo, 00015 Rome, Italy
3  Council for Agricultural Research and Economics, Research Centre for Agriculture and Environment, Sericulture Laboratory, 35143 Padua, Italy
4  Research Group in AgroICT and Precision Agriculture (GRAP), Agrotecnio CENTRA-Center, University of Lleida, E25198 Lleida, Spain
5  Department of Environment and Soil Science, University of Lleida, E25198 Lleida, Spain
*  Correspondence: domenico.giora@phd.unipd.it

**Abstract:** In recent decades there has been an increasing use of remotely sensed data for precision agricultural purposes. Sericulture, the activity of rearing silkworm (*Bombyx mori* L.) larvae to produce silk in the form of cocoons, is an agricultural practice that has rarely used remote sensing techniques but that could benefit from them. The aim of this work was to investigate the possibility of using satellite imaging in order to monitor leaf harvesting in mulberry (*Morus alba* L.) plants cultivated for feeding silkworms; additionally, quantitative parameters on silk cocoon production were related to the analyses on vegetation indices. Adopting PlanetScope satellite images, four *M. alba* fields were monitored from the beginning of the silkworm rearing season until its end in 2020 and 2021. The results of our work showed that a decrease in the multispectral vegetation indices in the mulberry plots due to leaf harvesting was correlated with the different parameters of silk cocoons spun by silkworm larvae; in particular, a decrease in the Normalized Difference Vegetation Index (NDVI) and Soil Adjusted Vegetation Index (SAVI) had high correlations with quantitative silk cocoon production parameters ($R^2$ values up to 0.56, $p < 0.05$). These results led us to the conclusion that precision agriculture can improve sericultural practice, offering interesting solutions for estimating the quantity of produced silk cocoons through the remote analysis of mulberry fields.

**Keywords:** *Morus alba* L.; *Bombyx mori* L.; silk; cocoon; precision agriculture; remote sensing; sericulture; vegetation index; satellite images

## 1. Introduction

Sericulture, or silk farming, is the rearing of silkworms (*Bombyx mori* L.) to produce silk in form of cocoons. Since *B. mori* is basically a monophagous insect, the amount of available mulberry (*Morus alba* L.) leaf is a major constraint on the maximum productive potential because silkworm larvae can be fed only with fresh mulberry leaves [1,2].

Based on a thorough literature review, we discovered an important knowledge gap: at the moment, there are only "rules of thumb" that relate the potential yields of mulberry leaves to the potential yield in cocoons, but these are mostly based on the experience of farmers. To the best of our knowledge, the only relevant work on this topic was presented by Lakshmanan in 2007 [3], who proposed an econometric analysis of the factors influencing silk production. Lakshmanan considered the cultivation of mulberry as a factor influencing silk production in the quoted article, but only in the introduction to the proposed models of binary variables in regard to adoption/non-adoption of new high-yielding varieties of

mulberry for leaf production and in terms of adoption/non-adoption of irrigation in the investigated mulberry fields.

In terms of precision agricultural approaches, remote sensing techniques have been successfully applied to monitor several herbaceous and perennial tree crops [4–12]. Nevertheless, an extensive literature review revealed that remote sensing has rarely been applied to mulberry cultivation and production. To the best of our knowledge, the only relevant study using remotely sensed data in mulberry cultivation is the work of Purohit et al. [13]. Remote sensing data were used in this study to find suitable locations for mulberry cultivation. Environmental models, different soil characteristics (e.g., pH, texture, or soil depth) and climate data were used to perform a large-scale land classification of India to identify potential areas for establishing new mulberry fields. Nevertheless, despite the fact that Purohit et al. [13] focused on mulberry cultivation, remote sensing data were not directly linked to mulberry monitoring.

When remote sensing techniques are applied to agriculture, the crop canopy is usually the target. Remote sensing cameras or sensors can be mounted on unmanned aerial vehicles (UAVs), drones, aircraft, or satellites. Currently, various constellations of satellites are available for precision agricultural purposes [14], with PlanetScope satellites serving as an important source of remote sensing data that can be used for crop monitoring. The PlanetScope mission is composed of approximately 130 nanosatellites orbiting at a height of about 470 kilometres and equipped with a multispectral camera that collects data in four spectral bands (RGB and NIR), with a spatial resolution of 3.0 m and a revisiting time of up to one day. This high spatial resolution enables the accurate detection of plant vigour and its changes even in small fields and orchards; the high temporal resolution, on the other hand, facilitates the fine characterization of the temporal evolution of vegetation. Several types of information can be derived from remotely sensed data, the most useful of which are those derived from the analysis of vegetation indices for precision agricultural purposes. Vegetation indices can be described as proxies that quantitatively represent the status of vegetation [15,16]. Different studies have demonstrated that, using vegetation indices from remotely sensed data, it is possible to estimate plant biomass [4,17–19], water stress [5,20–23], yield level [7,8,24–29], and product quality [28,29]; this approach has been applied in recent decades to both annual herbaceous crops [22,24,26,28,29] and perennial tree crops [4,5,7,8,10–12]. To the best of our knowledge, no study has considered the possibility of applying remote sensing techniques, by means of vegetation indices, to *M. alba* fields.

In this context, precision agricultural and remote sensing techniques can benefit sericulture. A potentially interesting application to sericulture is the possibility of collecting information, in the form of vegetation indices, about leaf production and leaf harvesting in mulberry fields from satellite images and using it to estimate parameters relevant to the silk market as they relate to cocoon production.

The main objective of this work was to evaluate the applicability of classical basic procedures for the analysis of remote sensing images collected with high-spatial-resolution cameras from PlanetScope satellites to monitor the biomass of mulberry fields cultivated to feed silkworms. Additionally, the work aimed to investigate the possible correlation between harvested biomass and quantitative parameters from cocoon production. The specific objectives of this work were (a) to analyse the correlations and regressions between vegetation indices derived from remote sensing data and cocoon quantitative production parameters and (b) to identify the best-performing vegetation indices for estimating cocoon yield.

## 2. Materials and Methods

In the present work, we retrieved the satellite images of four mulberry fields located in the Veneto Region (Italy) from PlanetScope satellites by georeferencing their borders and defining the temporal period of interest, corresponding to the *B. mori* rearing periods in 2020 and 2021. From the downloaded images we computed several vegetation indices

and studied their temporal development as a consequence of *M. alba* development and leaf harvesting operations. Finally, we related the parameters derived by the analysis of vegetation indices to economically relevant parameters that characterize cocoon production. In Figure 1, a schematic representation of the process is provided; further description of the procedure is provided in the following paragraphs of this section.

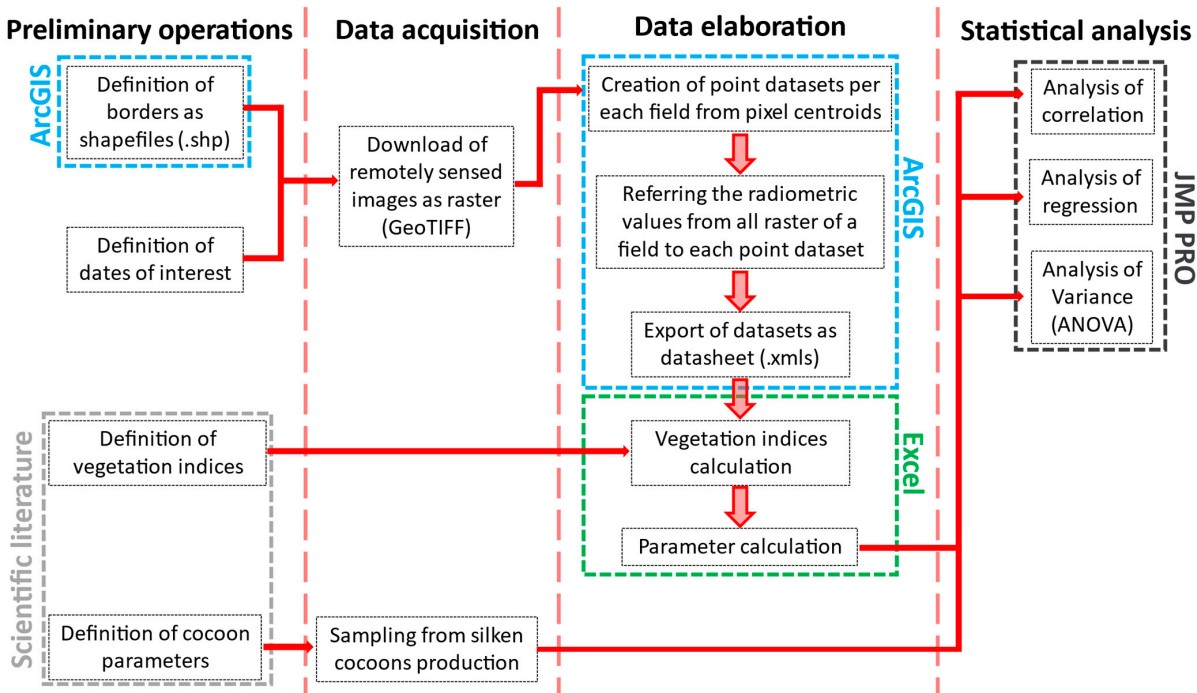

**Figure 1.** Schematic resume of working procedure.

## 2.1. Experimental Area Description

This study considered four *M. alba* fields used to feed silkworms for commercial silk production as the experimental areas. The leaves from these mulberry fields were harvested to feed *B. mori* larvae in two different years, namely 2020 and 2021. All of the selected farms and fields were located in the Veneto Region, north-eastern Italy, in the provinces of Padova and Treviso; Figure 2 depicts their geographic location.

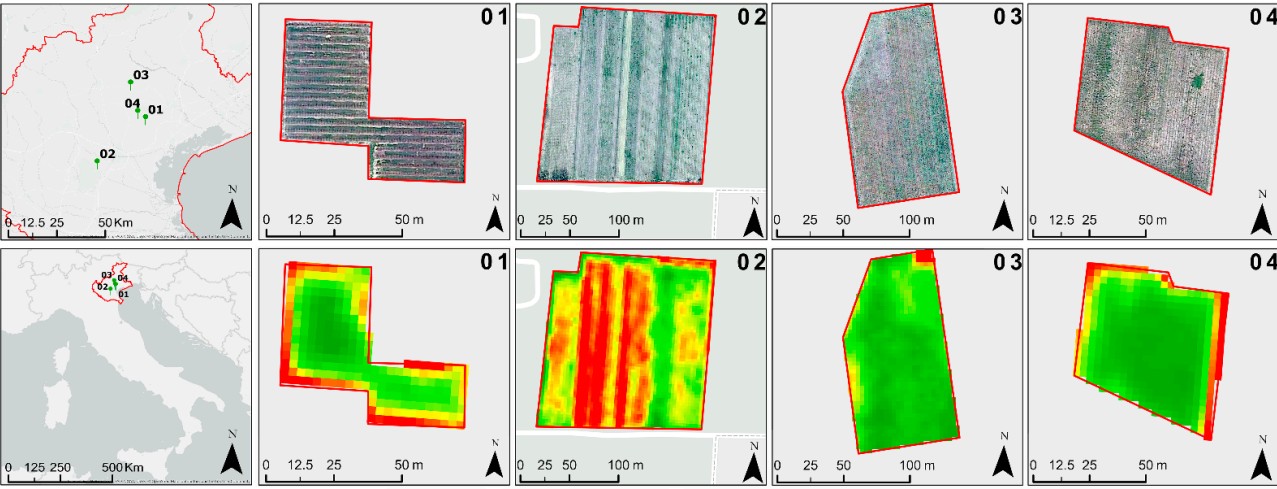

**Figure 2.** Location of *M. alba* experimental fields considered in the present work and their representation: top, high resolution from Google Earth (visible), bottom, high resolution from PlanetScope (NDVI).

The mulberry fields selected for this study were representative of typical Italian farms, with an orchard area smaller than 5 hectares (data from ISTAT, Italian National Statistical Institute, authors' elaboration).

In the Italian scenario, mulberry fields were typically arranged in two ways, as follows:

- High density: from 0.6 to 1.2 m in rows and 1.3 m between rows. This arrangement was represented by Field 02 and Field 04.
- Low density: 3.5 m in rows and 4 m between rows. This arrangement was represented by Field 01 and Field 03.

In both arrangements, plants at the maximum of vegetative growth tended to cover almost the entire inter-row space.

### 2.2. Silkworm Rearing

The life cycle of *B. mori*, from the hatching to the cocoon production period, is divided into five phases of active growth and alimentation called *instars*, which are interrupted by four phases of momentary halting of alimentation called *moultings*, which are characterised by important physiological activities. In this work, we referred to this simplified life cycle of *B. mori* as the "*commercial* life cycle" to distinguish it from the *biological* life cycle. A more detailed presentation of the biological life cycle of *B. mori* was beyond the scope of this article. For more details about the biology and physiology of the insect, see the Sericulture Training Manual [1] and the Encyclopedia of Insects [2].

All the *B. mori* polyhybrid larvae involved in the present work were reared from hatching to the beginning of the third instar in a centralized building and then distributed to the farmers. In fact, the first two instars of *B. mori* larvae are less resistant to diseases, suboptimal environmental conditions, and malnutrition [1,2].

Starting from the third instar, the *B. mori* larvae were reared by the farmers. Under standard rearing conditions [1], the third instar lasts four days, and the total consumption of fresh leaves per box (corresponding to 20,000 eggs) is approximately 13–15 kg. The fourth instar lasts five days, and the fresh leaf consumption during this period is approximately 10% of the total demand [2]. During the fifth instar, *B. mori* larvae ingest approximately the remaining 88% of the fresh leaves needed for their development [2], corresponding to approximately 300–340 kg of fresh leaves; this final instar lasts seven to eight days under standard conditions. The fifth instar is the period when the silk proteins for cocoons are actively biosynthesized by *B. mori* larvae and, therefore, farmers must provide an adequate supply of mulberry leaves. The last part of the commercial life cycle of *B. mori* is the cocoon production period, or mounting period, which lasts eight to ten days; during this period, the insects no longer need to be fed.

### 2.3. Data Acquisition and Analysis

2.3.1. Cocoon Yield Data Collection

To characterize the cocoon production of different farmers, five objective parameters were considered:

- *Total production:* This was the total production of cocoons from all reared boxes, per farmer and per year.
- *Production per box:* This was calculated as total production/number of boxes.
- *Average cocoons weight:* This was the average value of the individual weights of sampled cocoons.
- *Average silk shells weight:* This was the average value of the individual weights of sampled cocoons after the removal of the pupae.
- *Average silk percentage:* This was the average value of, calculated as silk shell weight/cocoon weight $\times$ 100.

The last three parameters were measured by sampling 30–40 cocoons per replicate, i.e., per farmer and per year; weights were measured as detailed in Saviane et al. [30] using a precision balance with a resolution of milligrams.

### 2.3.2. Remote Sensing Data

Remotely sensed data were used to track the evolution of the in-field situation of the mulberry fields according to the feeding necessities of silkworms in each farm. To this aim, data from the PlanetScope constellation (Planet Labs, Inc., San Francisco, CA, USA) were examined and downloaded.

To download the multispectral data from PlanetScope satellites, regions of interest (ROIs) were delineated for each mulberry field as a shapefile.

Rearing periods were defined on the basis of the record journals for silk production filled by the farmers themselves and in agreement with the official protocols and periods defined by the official national best practices. Key dates in these rearing periods were defined as follows:

- *Start of Feeding:* This day marked the start of harvesting of mulberry leaves in the fields. This day corresponded to the first day of the third instar when the *B. mori* larvae were given to farmers.
- *Start of the fourth instar:* The fourth larval instar of the silkworms began on this day.
- *Start of the fifth instar:* The last larval instar of the silkworms began on this day.
- *End of Feeding:* This was the last day of the fifth instar. The silkworm larvae stopped feeding on this day and began to spin cocoons.

Dates for both 2020 and 2021 and per field are listed in Table 1.

**Table 1.** Dates when PlanetScope images of each field were downloaded in relationship with each rearing period.

| Field ID | Spring 2020 | | | | Spring 2021 | | | |
|---|---|---|---|---|---|---|---|---|
| | Start of Feeding | Start of 4th Instar | Start of 5th Instar | End of Feeding | Start of Feeding | Start of 4th Instar | Start of 5th Instar | End of Feeding |
| 01 | 15 May | 20 May | 25 May | 2 June | 27 May | 1 June | 8 June | 15 June |
| 02 | 15 May | 21 May | 27 May | 2 June | 27 May | 1 June | 9 June | 13 June |
| 03 | 15 May | 20 May | 25 May | 2 June | 27 May | 1 June | 6 June | 13 June |
| 04 | 15 May | 20 May | 25 May | 2 June | 27 May | 1 June | 9 June | 17 June |

The downloaded PlanetScope images belonged to the type of surface reflectance (bottom of atmosphere reflectance), and only images with less than 30% cloud coverage were pre-selected. PlanetScope satellites were launched in different flocks and, according to this, they mounted a different type of sensor, namely PS2, PS2.SD, and PSB.SD [31]. Preferentially, we downloaded images from satellites equipped with PS2.SD or PSB.SD to reduce the source of error due to sensor differences; when this was not possible, PS2 images were also used. In this way, a total of 32 images were selected and downloaded from the platform.

### 2.4. Feature Extraction from Planet Data

Vegetation indices, as mentioned in the Introduction, are instruments designed to retrieve information about vegetation from remotely sensed images. The theoretical foundation of vegetation indices is that vegetation responds precisely to different wavebands. The absorption of light by leaves depends on their status: healthy leaves register a peak of light absorption, i.e., a low reflection, of incident radiation in the red (R) region of the spectrum and a high reflection of wavelengths in the near-infrared region (NIR), while senescent leaves reflect more red radiation than healthy ones. The peculiar pattern of light reflectance of leaves allows for the discrimination of vegetation from bare soils, water, or urbanized regions [16]. Considering remotely sensed images storing-per-pixel reflectance values in different wavelengths, vegetation indices are a mathematical instrument that combine information from different wavebands to discriminate and study vegetated areas, and they are generally based on the differences in red–NIR reflectance values. The simplest vegetation indices considered the ratio between the reflectance of NIR–red, being high

(25–30) in pixels representing dense and healthy vegetation. A step forward is made by normalizing the difference between NIR and red bands; normalized indices, as NDVI [32], are more stable [16] and simplify the interpretation and comparison of data since it ranges between ±1. Several authors have subsequently proposed modifications to NDVI in order to reduce the influence of the signals of soil, this being the SAVI [33] or its modifications, MSAVI2 [34], introducing corrective factors, and to reduce the influence of aerosols in the red band, in this case by correcting the red reflectance with information from reflectance in the blue region of spectrum.

In the present work, seven vegetation indices were calculated using the data organised in spreadsheets. For the present work, seven different vegetation indices were considered, selecting the most commonly used and quoted in the scientific literature for precision agricultural and crop monitoring purposes as well as the ones that could be computed from the spectral bands available from the PlanetScope images. Table 2 lists the used vegetation indices.

**Table 2.** Vegetation indices used in the present study.

| Index | Reference | Range |
|---|---|---|
| Atmospherically Resistant Vegetation Index (ARVI) | Kaufman and Tanré [35] | $(-1, +1)$ |
| Enhanced Vegetation Index (EVI) | Wang et al. [36] | $(-1, +1)$ |
| Green Normalized Difference Vegetation Index (GNDVI) | Wang et al. [37] | $(-1, +1)$ |
| Modified Soil Adjusted Vegetation Index (MSAVI2) | Qi et al. [34] | $(-1, +1)$ |
| Normalized Difference Vegetation Index (NDVI) | Rouse et al. [32] | $(-1, +1)$ |
| Soil Adjusted Vegetation Index (SAVI) | Huete [33] | $(-1, +1)$ |
| Visual Atmosphere Resistance Index (VARI) | Schneider et al. [38] | $(-1, +1)$ |

According to Huete [15], correcting $\gamma$ factor for ARVI was set as $\gamma = 1$ and correcting L factor for SAVI was set to $L = 0.5$.

Following the calculation of the vegetation indices, data were rearranged according to vegetation index and the chronological flow of the selected days. Table 2 shows the dates for which the vegetation indices were calculated (dates reported on the official traceability formats of silk production compiled by farmers). The total amount of information for the selected dates amounted to 1.4 Mb, with an average digitization footprint for the operation of 0.31 Mb/(ha·year) [39].

After the calculation of the vegetation indices, six parameters were extracted from this analysis. The mean values of each vegetation index, per field and per year, were computed in each date mentioned in Section 2.3.2, thus obtaining four average values per field and year of each vegetation index. Additionally, we computed the decrease in the vegetation indices during the rearing period of *B. mori,* subdividing them into two parts:

- *Decreasing during third and fourth instars:* The difference in values between the *start of the fifth instar* day and *start of the feeding* day was calculated. The authors determined the values corresponding to the first quartile, *quart1st*, by converting the resulting negative values (i.e., the points associated with a decrease in the selected vegetation index) into positive ones. The authors then calculated the average value $av_{tot}$ while excluding the values lower than *quart1st*, which were regarded as points with erratic variations in the vegetation index evolution. The measurement error was determined as the standard deviation of error, $SD_{err}$, from previously excluded values. This parameter thus represented the areas impacted by leaf harvesting during the third and fourth instars.

- *Decreasing during the fifth instar:* This parameter was calculated in the same way as the previous one but by determining the differences in values of the *end of feeding* and the *start of the fifth instar*; the same data filtering and correction were applied. As a result, this parameter took into account only the points that represented the areas impacted by leaf harvesting during the fifth instar.

### 2.5. Statistical Analysis

Two-way factorial ANOVA was performed on the data regarding the average cocoon weight, average silk shell weight, and average silk percentage with the four different mulberry fields and the two considered years as factors; the interaction between the two factors was also considered. When statistically significant differences were found, Tukey's HSD tests were performed as post hoc tests for mean separation. To analyse the data of the average silk percentage, an arcsine transformation was applied.

Correlation and regression analyses on the recorded data were performed. Correlations between the dataset of the cocoon and vegetation index parameters were analysed in depth. To this aim, the Pearson correlation coefficient *r* was calculated.

In addition, linear regression analyses between the cocoon and vegetation index parameters were performed with the ordinary least squares method; the statistical significance of defined regressions was tested using ANOVA.

For the statistical analysis of the data, JMP Pro Version 15.2.0 (JMP Statistical Discovery LLC, SAS Institute, Cary, NC, USA) statistical software and Microsoft® Excel® software were used.

## 3. Results and Discussion

In the present paragraph, we present the results of our work and provide a discussion about them. This paragraph is organized in four sub-sections, covering the results of the analysis of the quantitative parameter of cocoons (from Section 2.3.1), the temporal development of vegetation indices in the selected dates (from Sections 2.3.2 and 2.4), and the correlation and regression analysis (from Section 2.5).

In general, the present work analysed the possibility of estimating the yield of silk cocoons spun by the silkworm *B. mori* using different remotely sensed vegetation indices obtained from mulberry fields cultivated for insect feeding. Therefore, the present work applied to a rather original field an approach which has been successfully applied in the recent past to other tree crops through different vegetation [4,5,7,8,10–12,40,41]. In this work, the authors considered the average values of the vegetation indices on precise days and their evolution during the considered timespans.

### 3.1. Cocoon Parameters

Table 3 summarises total production and production per box; Table 4 presents the data on the average cocoon weight, silk shell weight, and silk percentage.

**Table 3.** Productive data per farm and per year.

| Field ID | Years | Number of Boxes | Total Production [kg] | Production per Box [kg/Box] |
|---|---|---|---|---|
| 01 | 2020 | 2 | 95.42 | 47.71 |
|    | 2021 | 2 | 92.77 | 46.39 |
| 02 | 2020 | 1 | 27.94 | 27.94 |
|    | 2021 | 1 | 33.51 | 33.51 |
| 03 | 2020 | 3 | 130.01 | 43.34 |
|    | 2021 | 4 | 153.21 | 38.30 |
| 04 | 2020 | 3 | 102.90 | 34.30 |
|    | 2021 | 1 | 35.40 | 35.40 |

The cocoon data were analysed by performing two-way factorial ANOVA. For the average cocoon weight, average silk shell weight, and average silk percentage, statistically significant differences ($p < 0.01$) were found. The factor "field" was statistically significant for each of the three parameters under consideration and similar results were obtained for the interactions; the factor "year" was statistically significant only in the case of the average silk percentage. Since the omnibus test evidenced statistically significant differences, we

performed the Tukey HSD test as a post hoc test for mean separation; the results of the Tukey HSD test ($\alpha$ = 0.05) are reported in Table 4.

**Table 4.** Average cocoon weight, average silk shell weight, and average silk percentage of sampled cocoons, divided per farm and per year. Different letters indicate significant differences among samples according to Tukey's HSD test ($\alpha$ = 0.05).

| Field ID | Years | Av. Cocoon Weight $\pm$ SD [g] | Av. Silk Shell Weight $\pm$ SD [g] | Av. Silk Ratio $\pm$ SD [%] |
|---|---|---|---|---|
| 01 | 2020 | 2.620 $\pm$ 0.379 a | 0.570 $\pm$ 0.075 a | 22.0 $\pm$ 1.8 a |
| | 2021 | 2.429 $\pm$ 0.367 ab | 0.538 $\pm$ 0.054 a | 22.5 $\pm$ 2.8 a |
| 02 | 2020 | 1.397 $\pm$ 0.221 d | 0.283 $\pm$ 0.050 d | 20.3 $\pm$ 1.4 b |
| | 2021 | 1.601 $\pm$ 0.279 d | 0.316 $\pm$ 0.055 d | 19.9 $\pm$ 2.6 b |
| 03 | 2020 | 2.389 $\pm$ 0.330 ab | 0.487 $\pm$ 0.067 b | 20.4 $\pm$ 1.3 b |
| | 2021 | 2.347 $\pm$ 0.395 b | 0.538 $\pm$ 0.078 a | 23.2 $\pm$ 3.1 a |
| 04 | 2020 | 2.065 $\pm$ 0.302 c | 0.413 $\pm$ 0.047 c | 20.2 $\pm$ 2.0 b |
| | 2021 | 1.905 $\pm$ 0.267 c | 0.408 $\pm$ 0.045 c | 21.7 $\pm$ 2.6 ab |

### 3.2. Evolution of Vegetation Indices

Except for EVI, the mean values of the examined vegetation indices increased in the first half of *B. mori*'s rearing season, increasing from the *start of feeding* to the *start of the fifth instar*, and from the *start of the fifth instar* to the *end of feeding*, while the mean values of the vegetation indices decreased. The rearing seasons of 2020 and 2021 both supported this tendency (Figure 3). In Figure 4, a graphical example of the in-field evolution of vegetation is provided.

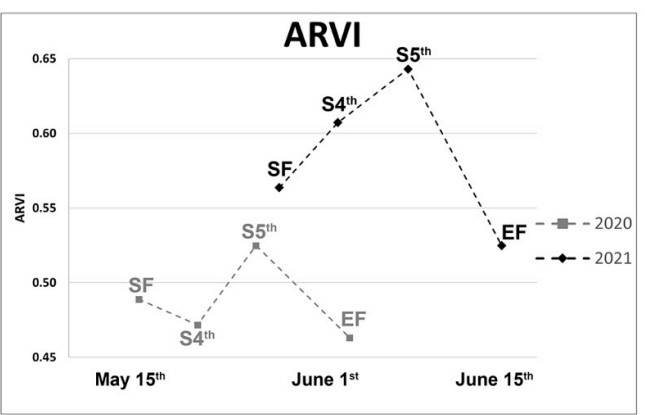
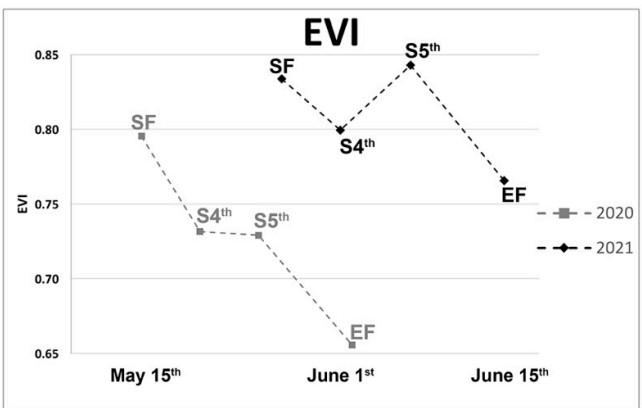
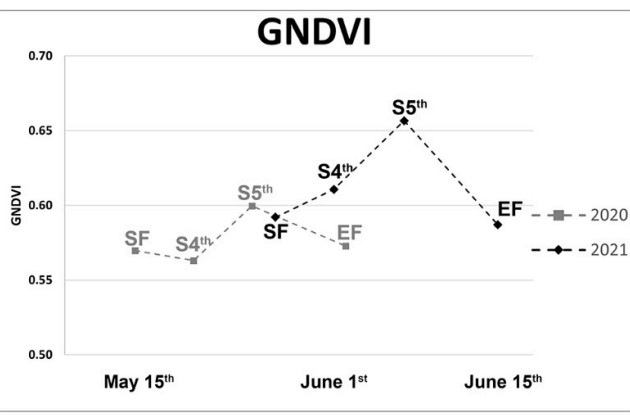
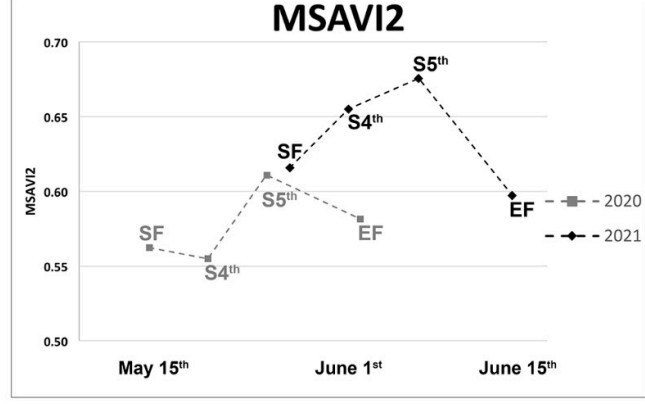

**Figure 3.** *Cont.*

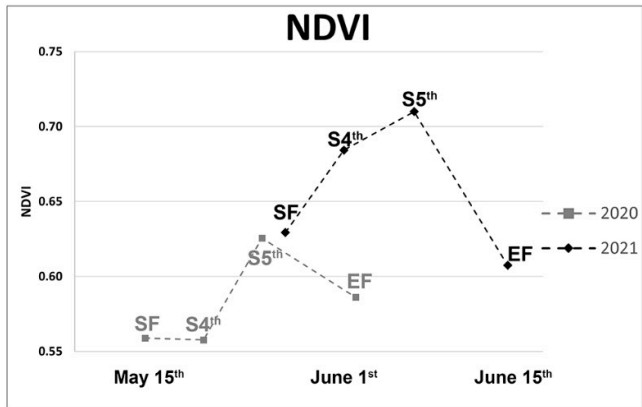
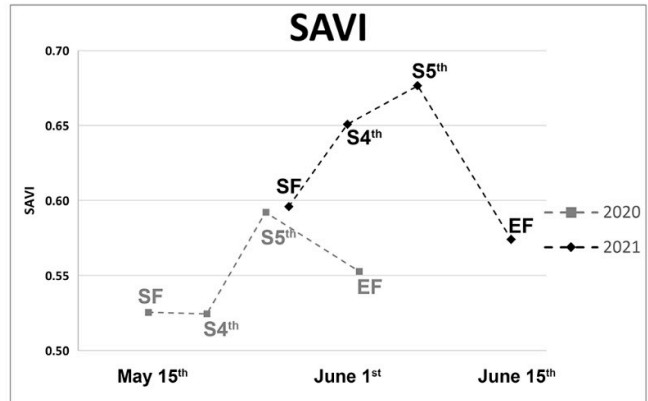

**Figure 3.** Evolution of five vegetation indices from the considered *M. alba* fields on the four previously defined dates: SF = start of feeding, S4th = start of fourth instar, S5th = start of fifth instar, EF = end of feeding.

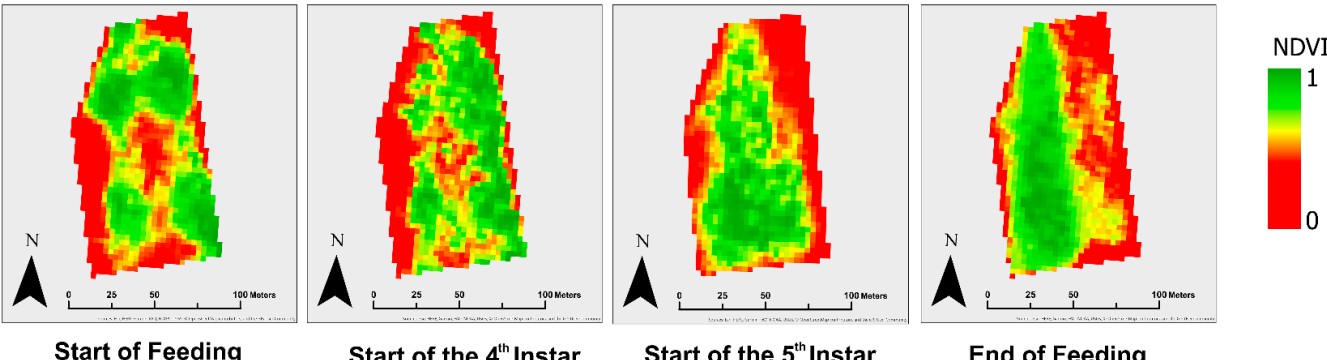

**Figure 4.** Spatial evolution of NDVI from the ID 03 *M. alba* field during 2021 rearing season.

The graphs in Figure 3 show that the entire rearing season was about 12 days later in 2021 than in 2020, resulting in higher mean values of the vegetation indices (see Table 2). Additionally, compared to 2020, the changes in the mean values of the vegetation indices were greater in 2021. Analysing the time series of the vegetation indices remotely collected from crops by satellites is a well-established procedure in the field of remote sensing and precision agriculture, as it allows researchers to follow the growing seasons of crops and derive and analyse several biophysical parameters from them, such as the phenological stage of monitored crops [27,40–42]. Although similar methodologies could also have been tested for *M. alba*, at present no study has applied remote sensing techniques to *M. alba* cultivation. This first result of our works showed that this practice was also suitable for the characterization of *M. alba* fields, in particular the decrease in vegetation due to harvesting could be clearly detected during the fifth instar of *B. mori* larvae, thus opening new possible and more deep investigations into this topic.

In Table 5, NDVI is shown as an example of the extent of the vegetation index decreasing due to leaf harvesting; data are divided per field and per year in the table. As a first consideration, the magnitude of the decrease was higher in 2021 than in 2020 for all the fields; secondly, the decrease during the fifth instar was greater than the total in the third and fourth instars summed together.

In general, the standard error $SD_{err}$ was low, except for Field 01 in both 2020 and 2021 and in Field 04 in 2021. In some cases, as for Field 01 in 2021, the decrease in the vegetation indices was not detectable, and for this reason, this parameter was not considered in further analysis. A possible explanation for this is that the *B. mori* larvae consumed low quantities of leaf in this period [1,2]; thus, the effect of leaf harvesting on the vegetation indices was probably masked by the development of the plants, demonstrated by the general increasing

trends of the vegetation indices in the first part of the rearing seasons, as previously shown in Figure 3.

**Table 5.** Decrease in NDVI during 2020 and 2021 rearing seasons. Values of the decrease during the third and fourth instars and during the fifth instar are illustrated.

| Field ID | Years | Third + Fourth Instars | | | Fifth Instar | | |
|---|---|---|---|---|---|---|---|
| | | quart1st | $av_{tot}$ | $SD_{err}$ | quart1st | $av_{tot}$ | $SD_{err}$ |
| 01 | 2020 | 0.005 | 0.024 | 0.001 | 0.074 | 0.145 | 0.022 |
| | 2021 | <0.001 | <0.001 | <0.001 | 0.152 | 0.233 | 0.038 |
| 02 | 2020 | 0.009 | 0.028 | 0.003 | 0.015 | 0.041 | 0.004 |
| | 2021 | 0.007 | 0.020 | 0.002 | 0.024 | 0.048 | 0.006 |
| 03 | 2020 | 0.015 | 0.032 | <0.001 | 0.013 | 0.036 | 0.004 |
| | 2021 | 0.004 | 0.018 | 0.001 | 0.089 | 0.117 | 0.007 |
| 04 | 2020 | 0.010 | 0.048 | 0.003 | 0.039 | 0.086 | 0.011 |
| | 2021 | 0.022 | 0.059 | 0.015 | 0.045 | 0.101 | 0.031 |

*3.3. Correlation Analysis between Vegetation Indices and Cocoon Production Parameters*

The values of Pearson's correlation coefficient *r* between the cocoon production data and vegetation indices are reported in Table 6 along with the correlations' statistical significance; to better highlight interesting correlations, statistical significances at different levels are highlighted with different colours.

The first four columns on the left-hand side of the matrix (Table 6) illustrate the correlation coefficients among the parameters derived only from cocoons. The total production was highly correlated with the average weight of both the cocoons and shells, with both being statistically significant; when the production per box was taken into account, the correlations were even higher, and the statistical significance increased. The correlations among these two parameters and the average silk percentage were not statistically significant. As expected, the average weight of the cocoons was highly correlated with the average weight of the silk shells ($r = 0.98$); these results were in accordance with previous studies on the topic [43–46].

The columns in the right-hand side of the matrix show the correlation coefficients between the vegetation indices and the considered parameters of cocoon production. The table is arranged as follows: the main row from the left indicates the cocoon production parameter that was tested against a particular parameter derived from the vegetation indices. So, for example, considering the production parameter "Cocoon weight" in the left side of the table, "0.69 *" is the value of the correlation coefficient *r* obtained testing the correlation of the "decreasing during the fifth instar" of "NDVI" against the "Cocoon weight" itself. The average values of the vegetation indices on specific days were not correlated with the cocoon production parameters; only the decrease in the chosen vegetation indices during the fifth instar was correlated with some of the considered cocoon production parameters.

More specifically, no statistically significant correlation between the vegetation indices and the total production of cocoons was found. This was explained by the fact that, as shown in Table 1, the number of boxes reared by each farmer changed over the years, and, accordingly, the total production did so too. Therefore, the correlations with increased indices normalized the total production per number of boxes (i.e., the production per box parameter): good results were achieved by decreasing the ARVI ($r = 0.73$), MSAVI2 ($r = 0.71$), NDVI ($r = 0.74$), and SAVI ($r = 0.74$). In addition, the average weight of the cocoon, the average weight of the silk shell, and the average silk percentage were correlated with a decrease in the considered vegetation indices at different levels of statistical significance. In those last cases, the best results were achieved by decreasing the NDVI tested against the silk shell weight ($r = 0.75$) and decreasing the SAVI against the silk percentage ($r = 0.75$).

**Table 6.** Pearson's correlation coefficients *r* describing correlations among vegetation indices and cocoon production data.

| Parameter | Production per Box | Cocoons Weight | Silk Shell Weight | Silk Ratio | Date | ARVI | EVI | GNDVI | MSAVI2 | NDVI | SAVI | VARI |
|---|---|---|---|---|---|---|---|---|---|---|---|---|
| Total production | 0.57 ns | 0.80 ** | 0.78 ** | 0.49 ns | Start of feeding | 0.15 | −0.13 | 0.27 | 0.19 | 0.18 | 0.18 | −0.17 |
| | | | | | Start of the 4th instar | 0.35 | 0.31 | 0.44 | 0.39 | 0.37 | 0.37 | 0.12 |
| | | | | | Start of the 5th instar | 0.51 | 0.36 | 0.54 | 0.54 | 0.54 | 0.54 | 0.44 |
| | | | | | End of feeding | 0.38 | 0.32 | 0.41 | 0.38 | 0.38 | 0.38 | 0.20 |
| | | | | | Decreasing during the 5th instar | 0.25 | −0.10 | 0.10 | 0.19 | 0.34 | 0.32 | −0.12 |
| Production per box | - | 0.93 *** | 0.90 *** | 0.55 ns | Start of feeding | −0.13 | 0.01 | −0.08 | −0.10 | −0.13 | −0.13 | −0.26 |
| | | | | | Start of the 4th instar | 0.11 | 0.11 | 0.13 | 0.18 | 0.13 | 0.13 | 0.01 |
| | | | | | Start of the 5th instar | 0.23 | 0.21 | 0.36 | 0.24 | 0.19 | 0.19 | −0.16 |
| | | | | | End of feeding | −0.03 | −0.05 | −0.10 | −0.04 | −0.05 | −0.05 | 0.02 |
| | | | | | Decreasing during the 5th instar | 0.73 ** | 0.49 | 0.68 * | 0.71 ** | 0.74 ** | 0.74 ** | −0.05 |
| Cocoons weight | - | - | 0.98 *** | 0.64 * | Start of feeding | 0.05 | −0.06 | 0.16 | 0.09 | 0.07 | 0.07 | −0.23 |
| | | | | | Start of the 4th instar | 0.26 | 0.27 | 0.33 | 0.32 | 0.27 | 0.27 | 0.06 |
| | | | | | Start of the 5th instar | 0.42 | 0.32 | 0.55 | 0.44 | 0.41 | 0.41 | 0.09 |
| | | | | | End of feeding | 0.19 | 0.14 | 0.14 | 0.18 | 0.18 | 0.18 | 0.19 |
| | | | | | Decreasing during the 5th instar | 0.65 * | 0.34 | 0.55 | 0.6 | 0.69 * | 0.68 * | −0.08 |
| Silk Shell weight | - | - | - | 0.77 ** | Start of feeding | 0.05 | −0.12 | 0.14 | 0.09 | 0.08 | 0.08 | −0.14 |
| | | | | | Start of the 4th instar | 0.27 | 0.29 | 0.29 | 0.31 | 0.27 | 0.27 | 0.12 |
| | | | | | Start of the 5th instar | 0.40 | 0.29 | 0.57 | 0.43 | 0.41 | 0.40 | 0.06 |
| | | | | | End of feeding | 0.15 | 0.13 | 0.10 | 0.13 | 0.13 | 0.13 | 0.13 |
| | | | | | Decreasing during the 5th instar | 0.68 * | 0.30 | 0.61 | 0.66 * | 0.75 ** | 0.74 ** | −0.18 |
| Silk Percentage | - | - | - | - | Start of feeding | 0.12 | −0.17 | 0.11 | 0.14 | 0.15 | 0.15 | 0.24 |
| | | | | | Start of the 4th instar | 0.27 | 0.35 | 0.16 | 0.23 | 0.24 | 0.24 | 0.37 |
| | | | | | Start of the 5th instar | 0.32 | 0.15 | 0.55 | 0.34 | 0.35 | 0.35 | 0.03 |
| | | | | | End of feeding | 0.06 | 0.16 | <0.01 | −0.02 | <0.01 | <0.01 | <0.01 |
| | | | | | Decreasing during the 5th instar | 0.61 | 0.10 | 0.63 * | 0.68 * | 0.74 ** | 0.75 ** | −0.35 |

*** statistical significance at α = 0.01, ** statistical significance at α = 0.05, * statistical significance at α = 0.1.

In general, not all the analysed vegetation indices could be used to estimate the parameters related to silk cocoon production. According to [15], in fact, not every vegetation index works in the same way when applied to different crops, and researchers should perform comparative analyses to find the best-performing vegetation indices for their field of study. In this case, we found that the ARVI, NDVI, and SAVI performed better than the other indices. Two vegetation indices, EVI and VARI, did not show any correlation with the considered cocoon production parameters

Lastly, we found high correlations among the different indices (data not shown). This behaviour was caused by the fact that all of the chosen vegetation indices were calculated through the same spectral bands. These results clearly indicated that a multiple regression analysis with these vegetation indices should be avoided due to bias related to the collinearity of variables.

### 3.4. Regression Analysis of Vegetation Indices and Silk Production Parameters

In the following paragraphs, the most relevant results for the tested regressions and their statistical significance are reported. This section is organised into paragraphs for the different analysed silk cocoon production parameters to make the discussed themes easier to understand.

### 3.4.1. Production per Box

The determination coefficient $R^2$ values of the NDVI and SAVI were approximately 0.55, and for the ARVI, it was approximately 0.52; in all three cases, the regressions were statistically significant with associated *p*-values lower than 0.05. The regressions between the ARVI, NDVI, and SAVI and the average production per box are shown in Figure 5.

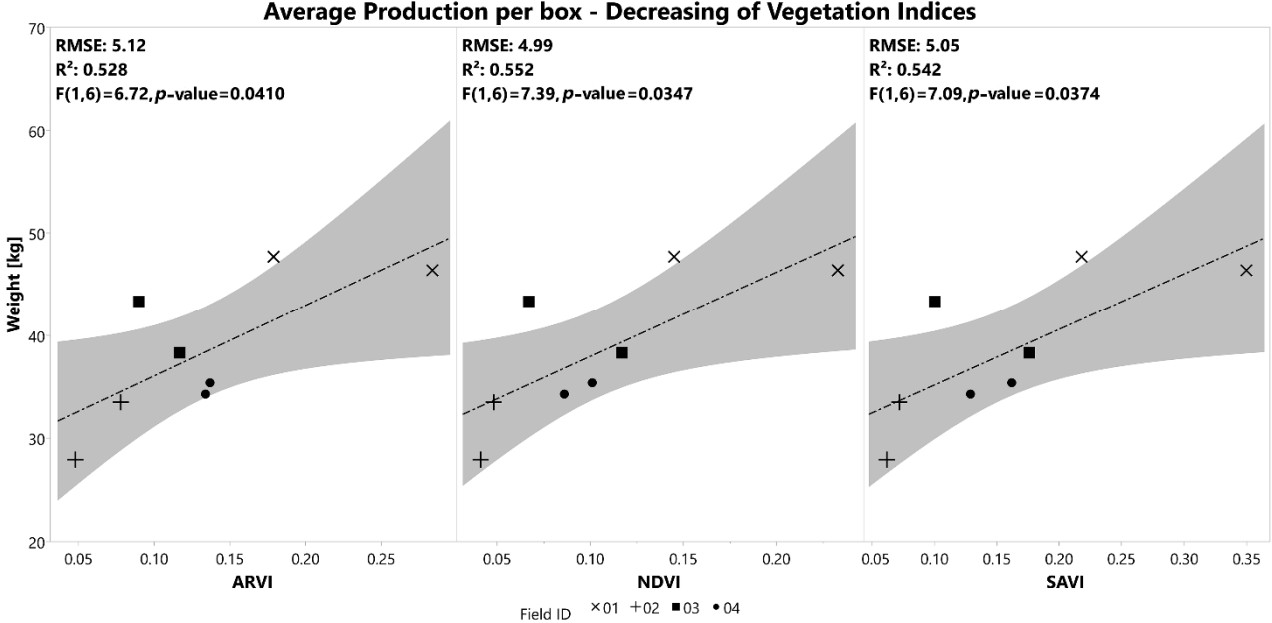

**Figure 5.** Average production per box [kg] in relation to decreasing vegetation indices during the fifth instar of *B. mori*. Gray shaded areas represent the confidence interval of regression at 95%.

### 3.4.2. Average Weight of Cocoons

With values of the determination coefficient $R^2$ of approximately 0.47 for the NDVI and SAVI and approximately 0.42 for the ARVI, the accuracy of the calculation of the total weight of cocoons using the vegetation indices was low. The regressions with all three considered indices exhibited a low statistical significance. The regressions between the ARVI, NDVI, and SAVI and the average weights of cocoons are shown in Figure 6.

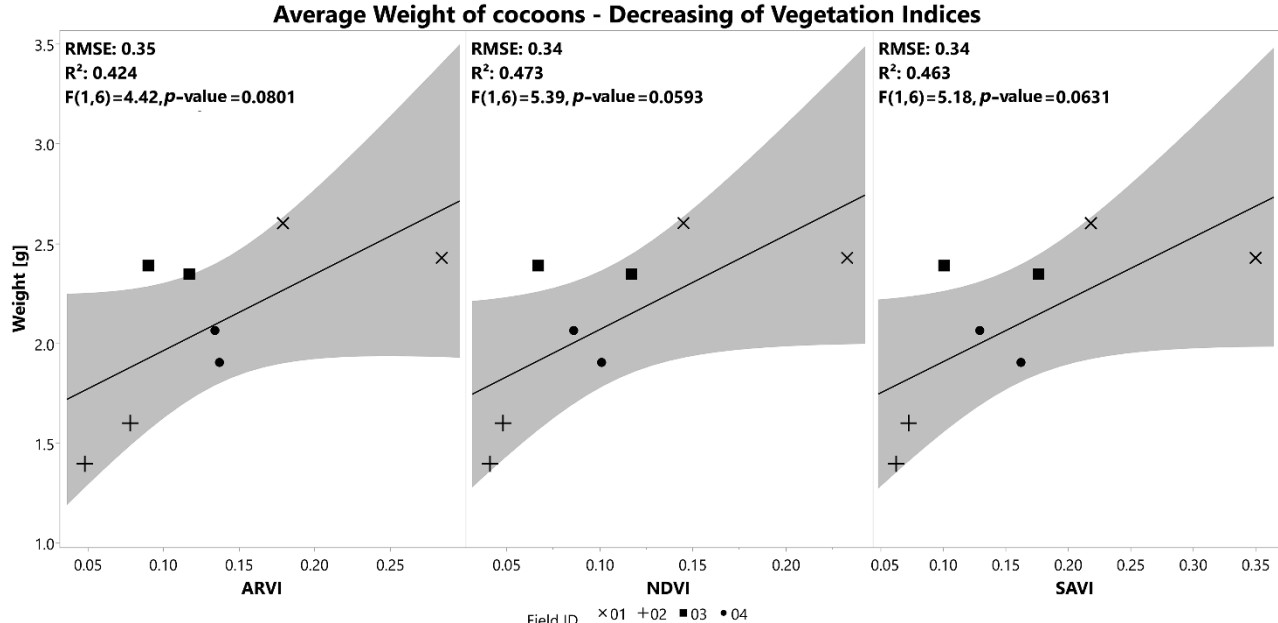

**Figure 6.** Average weight of cocoons [g] in relation to decreasing vegetation indices during the fifth instar of *B. mori*. Gray shaded areas represent the confidence interval of regression at 95%.

### 3.4.3. Average Weight of Silk Shell

The determination coefficient $R^2$ values computed for the best-fitting regression lines were approximately 0.56 for the NDVI and SAVI. The regressions with decreasing NDVI and SAVI were statistically significant with associated *p*-values lower than 0.05. The regressions between the ARVI, NDVI, and SAVI and the average weights of the silk shells are shown in Figure 7. This result was particularly interesting since the weight of silk shells is one of the most important parameters used for the estimation of yield in raw silk terms [1,42,43].

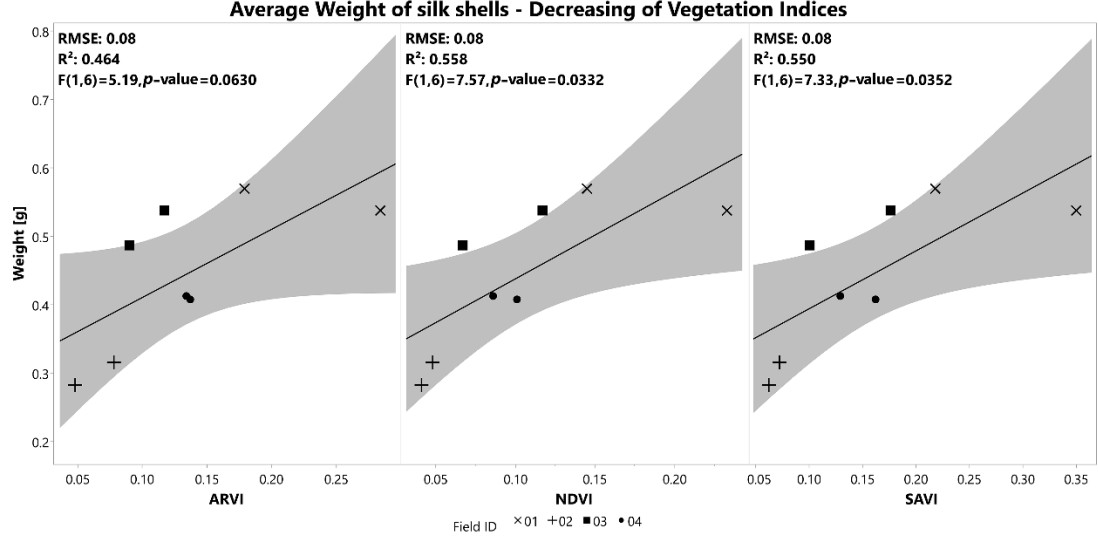

**Figure 7.** Average weight of silk shell [g] regressions in relation to decreasing vegetation indices during the fifth instar of *B. mori*. Gray shaded areas represent the confidence interval of regression at 95%.

### 3.4.4. Average Silk Percentage

Among all the considered indices, for the NDVI and SAVI, $R^2 = 0.56$; the regressions were statistically significant. The regression between the average silk percentage and the ARVI was low, with $R^2 = 0.37$, and was not statistically significant. In Figure 8, the graphs of the tested correlations are shown.

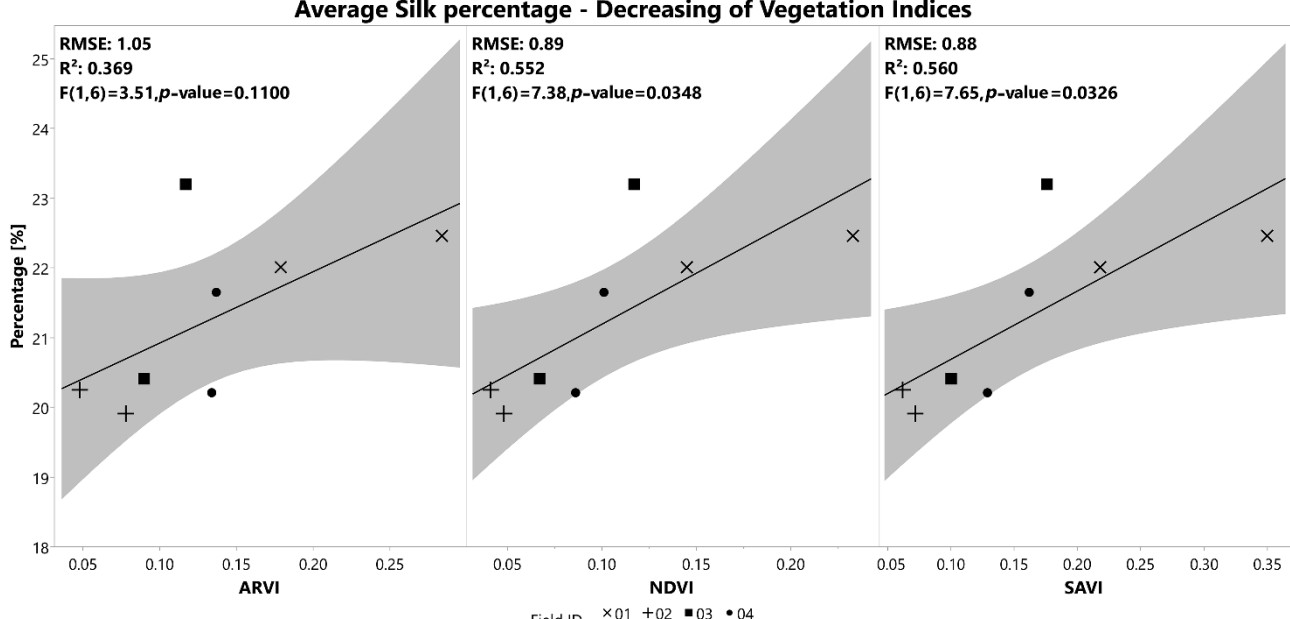

**Figure 8.** Average silk percentage of cocoons [%] regressions in relation to decreasing vegetation indices during the fifth instar of *B. mori*. Gray shaded areas represent the confidence interval of regression at 95%.

## 4. Conclusions

This research represents a first attempt at the possibility of applying precision agriculture to sericulture for estimating silk cocoon production, taking advantage of high-resolution remote sensing data. The reported results gave evidence of how a remote sensing approach could be usefully implemented for sericultural purposes, showing how economically relevant parameters, such as the silk shell weight of cocoons, were correlated with the temporal evolution of different vegetation indices. More specifically, the decrease in NDVI ($r = 0.75$) and SAVI ($r = 0.74$) over time were reported. Regression models were derived from these correlations, thus allowing for the estimation of the quoted parameters.

Additionally, this work opens up to the possibility of focusing on further well-established approaches typical of precision agriculture, such as the integration of different data for homogeneous management zone delineation, which here was meant to differentiate the harvested leaves for silkworms' feeding. Accordingly, future work will focus on the investigation of other correlations among relevant parameters characterizing leaf production (yield and quality) and remotely sensed vegetation indices.

**Author Contributions:** Conceptualization, J.A.M.-C. and D.G.; methodology, J.A.M.-C. and D.G.; software, D.G.; validation, S.C., A.S. and F.M.; formal analysis, D.G.; investigation, D.G.; resources, D.G.; data curation, D.G.; writing—original draft preparation, D.G.; writing—review and editing, S.C., J.A.M.-C., A.S., A.A., F.M. and L.S.; visualization, D.G., J.A.M.-C. and F.M.; supervision, J.A.M.-C.; project administration, L.S.; funding acquisition, L.S., A.A. and S.C. All authors have read and agreed to the published version of the manuscript.

**Funding:** This research was funded by the Veneto Region, Measure 16.1-2 Programme of Rural Development for the Veneto Region, 2014-2020-DGR 2175 del 23/12/2016, grant number: "Decree n. 55 of 4th December 2017 of financing of the project Serinnovation.".

**Data Availability Statement:** Data presented in this study are available on request from the corresponding author. The data are not publicly available because they can be used by the Operational Group "Serinnovation" for future economic applications.

**Acknowledgments:** The authors thank Planet Labs Inc. for the use of PlanetScope images under the Educational and Research License agreement. The authors thank the farmers of the Operational group "Serinnovation" and the farmers of "Bachicoltura setica" network for allowing to collect data from their farms. D.G. thanks the Erasmus+ mobility program and the GINI association, which allowed him carry out the research by his stay period in the University of Lleida (Spain).

**Conflicts of Interest:** The authors declare no conflict of interest. The funders had no role in the design of the study; in the collection, analyses, or interpretation of data; in the writing of the manuscript, or in the decision to publish the results.

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
