# Peer review of "Remote Sensing Imaging as a Tool to Support Mulberry Cultivation for Silk Production"

_remotesensing, doi:10.3390/rs14215450_

Round 1
Reviewer 1 Report (Previous Reviewer 1)
- L31. I would avoid saying "has proven" at this stage. If you must do it, the conclusions could be a better place.
- L93. Planet data is considered High spatial resolution, and is actually on the edge with Very High spatial Resolution data. Please remove the "medium" word.
- Figure 1. Could you add a short time series for one of the fields in order to better showcase how the use of Planet data at 1 day temporal resolution and 3m spatial one does work on the analysis that you are carrying out?.
- L173-180. Who defines these dates? Could you please add them in here?. I think the suggested time series from my previous comment can better fit in here. You should also add the exact dates that you had used for each/all fields. Since they are just 4, it will surely not be a problem.
- Section 2.3. This is not a GIS journal, but a Remote Sensing one. The first paragraph of this section is not required in the way you are currently presenting it. Unless you have actually performed some processing on the real data (which I know you have not since the data is ready to use from Planet), you do not require it.
- Table 2 is appearing too late, please move earlier. Information right after Table 2 seems to be a repetition. Please combine with the information added in lines 173-180.
- Figure 2 is appearing way too late. Onlly now I realized you were presenting what you call your method. General description of the "proposed/presented" method/methodology must be always added at the very beginning of the section. It must be acompanied by a short description of each step in order to help the possible reader(s) to understand where they are and where are they going. Please move to the right place.
- Section 3 should be presented in the same order as the methods' steps. You need to keep coherence on the organization. The general organization is now better, specially while having joined the discussion in here. But you need to be in line with things presented in the methodology. There are way too many sections in here, and it is difficult to follow the ideas. A small introduccion describing the general organization of this section is also missing. Please reduce some parts and be more direct with discussions. Avoid repetitions of information and keep some explanations in a general way. Make better use of the different figures/tables that you have added.
- L457. Please avoid the use of the word "pioneer". You are making it sound as if no-one had ever ever used any of the techniques you have presented in here. Whereas they have been used since ages, specially foro precision agriculture.
- Conclusions: Please reduce the lenght of conclusions and avoid stating possible future works that are not possiblel with the Planet data, given the lack of proper spectral bands.
Author Response
Dear Reviewer 1,
Thank you for your precise and detailed comments.
All of the reported positions of the modifications refer to the line numbers in the new manuscript with the "Track Changes" function activated.
L31. I would avoid saying "has proven" at this stage. If you must do it, the conclusions could be a better place. |
We have modified accordingly. Please see the abstract at line 31 |
- L93. Planet data is considered High spatial resolution and is actually on the edge with Very High spatial Resolution data. Please remove the "medium" word. |
Thank you, we have deleted the word “medium” even in other part where were present. |
- Figure 1. Could you add a short time series for one of the fields in order to better showcase how the use of Planet data at 1-day temporal resolution and 3m spatial one does work on the analysis that you are carrying out? |
We added a new image (Figure 4) that graphically represents, as an example, the evolution of NDVI over selected dates. |
- L173-180. Who defines these dates? Could you please add them in here? I think the suggested time series from my previous comment can better fit in here. You should also add the exact dates that you had used for each/all fields. Since they are just 4, it will surely not be a problem. |
As explained in section 2.1.2, the development of silkworms is discontinuous, since their growth passes through five active growing phases, called instar, interspersed with four non-growing phases, called moultings; moultings are important since they prepare larvae to the following instar, allowing the continuation of growing. The dates here considered refer to the date of moulting, that are registered by farmers in apposite forms used for the traceability of product and further analysis. |
- Section 2.3. This is not a GIS journal, but a Remote Sensing one. The first paragraph of this section is not required in the way you are currently presenting it. Unless you have actually performed some processing on the real data (which I know you have not since the data is ready to use from Planet), you do not require it. |
We have deleted this paragraph |
- Table 2 is appearing too late, please move earlier. Information right after Table 2 seems to be a repetition. Please combine with the information added in lines 173-180. |
We have moved the Table 2 in Section 2.2.2; now is Table 1. Additionally, we combine the information and now you can find the new arrangement of descriptions in lines 258 to 275 |
- Figure 2 is appearing way too late. Only now I realized you were presenting what you call your method. General description of the "proposed/presented" method/methodology must be always added at the very beginning of the section. It must be accompanied by a short description of each step-in order to help the possible reader(s) to understand where they are and where are they going. Please move to the right place. |
We moved the previous Figure 2 at the very beginning of the Section 2 and now is named as Figure 1. Additionally, we improved this section by adding a small paragraph describing the general procedure as from indications. Please, check the changes from line 103 to 111 |
- Section 3 should be presented in the same order as the methods' steps. You need to keep coherence on the organization. The general organization is now better, specially while having joined the discussion in here. But you need to be in line with things presented in the methodology. There are way too many sections in here, and it is difficult to follow the ideas. A small introduction describing the general organization of this section is also missing. Please reduce some parts and be more direct with discussions. Avoid repetitions of information and keep some explanations in a general way. Make better use of the different figures/tables that you have added. |
Thank you for the precise comment. First of all, we added a brief introduction to the section, and you can find it from line 310 to 314, trying to link the proposed methodology to results. Then, to follow the organization of the methodology section, we have moved the paragraph 3.2 (describing analysis of cocoon parameter) to 3.1; this section is then followed by the analysis of time series of VI, according to the description of dates in section 2.2.2. We preferred to keep separated paragraphs 3.3 and 3.4 to make easier the comprehension by the separation of correlation and regression analysis, but if for the referee is better to join them, we can handle that. Additionally, we condensate the discussion of results. |
- L457. Please avoid the use of the word "pioneer". You are making it sound as if no-one had ever used any of the techniques you have presented in here. Whereas they have been used since ages, specially for precision agriculture. |
Thank you for the comment. We have revised the text according to the suggestion. |
- Conclusions: Please reduce the length of conclusions and avoid stating possible future works that are not possible with the Planet data, given the lack of proper spectral bands. |
We have reduced the Conclusion section. |
Reviewer 2 Report (Previous Reviewer 2)
This paper examined the statistical modeling of silk production using PS images, which is a reasonable approach but requires some modification regarding the VI descriptions and modeling methods.
Line 202. More in-depth descriptions are necessary for the VI introductions, including the fundamental of the VI, the combination of bands, the range of the VI, and the meaning of the VI.
Line 401. Was the number of samples used in the regression model just eight? If so, n=8 is insufficient to discuss the statistical significance of the model.
Author Response
Dear Reviewer 2
Thank you for your precise and detailed comments.
All of the reported positions of the modifications refer to the line numbers in the new manuscript with the "Track Changes" function activated.
Line 202. More in-depth descriptions are necessary for the VI introductions, including the fundamental of the VI, the combination of bands, the range of the VI, and the meaning of the VI. |
We moved a part of the discussion upon vegetation indices from the paragraph Introduction to Paragraph 2.3. Here, we provided a more detailed discussion upon Vis (lines 221 to 241). We also improved Table 2 and corrected the actual Figure 3 where we found an error produced during graphs editing/plotting: thus scalebars of three graphs have been properly edited |
Line 401. Was the number of samples used in the regression model just eight? If so, n=8 is insufficient to discuss the statistical significance of the model. |
Thank you for the comment. We are aware that n=8 might be considered a poor dataset for a regression model; on the other hand, in our experience such number represents an acceptable effort for the farmer (see the comment on digitization footprint) and is in agreement with the common number of available dates for satellite images. Also, it should be considered that each value derived from both vegetation indices and cocoons parameters is the average of several data: - 30/40 data per cocoon parameter, as detailed in section 2.2.1 - at least 25/30 pixels for the decreasing of VI. Finally, we believe that the relevance of the contribution of this work is not on the development of predictive models (which might have limited geographical and variety validity) but on the introduction of a (new) approach (implementing remote sensing in sericulture) which might be replicated at farm or at consortium level. A comment has been added to make this point more clear. |
Round 2
Reviewer 1 Report (Previous Reviewer 1)
- Pay attention to the right use of template. L108.
- Section 2.3. title should be "Feature extraction from Planet data". You are not talking about any pre-processing step in here.
- L239. It is sub-section and not paragraph.
- Second paragraph of the conclusions must be deleted and replaced by a proper one describing possible future works. Further improve the current first paragraph by better describing the findings of your work.
Author Response
Dear Reviewer,
Thank you for your precise comments.
- Pay attention to the right use of template. L108. - Section 2.3. title should be "Feature extraction from Planet data". You are not talking about any pre-processing step in here. - L239. It is sub-section and not paragraph. |
Thank you, we have corrected the issues and changed the title of Section 2.3 according to the suggestion. |
Second paragraph of the conclusions must be deleted and replaced by a proper one describing possible future works. Further improve the current first paragraph by better describing the findings of your work. |
Thank you, we have now improved the conclusions section. In particular, we have added a brief discussion on the potential future research on the topic and improved the presentation of our results. |
This manuscript is a resubmission of an earlier submission. The following is a list of the peer review reports and author responses from that submission.
Round 1
Reviewer 1 Report
This manuscript presents the anaylisis carried out from studying silk production processes with different radiometric indices extracted from Planet data over some fields (with areas around 5ha) located in Italy. The manuscript itself does not provide a particular novelty, other than testing some new sort of data that has not been reported before in literature. Authors are not clear about the methodology itself, the results are way too long for the actual proposed method, and the discussion does not show that authors have been able to achieve their original goal. It is not clear to me what is the exact point of this manuscript. Title is misleading and must be changed.
ABSTRACT
- The abstract is generally well-written, but the novelty of the work is missing in here. It is not quiet clear what exactly are authors proposing. Please clarify and make sure to format the text in the right way.
INTRODUCTION
- L84. Please avoid the use of "believe" that is not a scientific expression.
- Generally well-written and now you do mention what you are proposing. Please make sure that you transfer this information to the abstract as well.
MATERIALS AND METHODS
- Figure 1. On the title: Is experimental and not esperimental. I assume the images shown in here are from VHR sensors. It would be good, and better, to show an actual Planet image in order to allow the possible reader to better understand how much details can you see at 3m spatial resolution.
- Section 2.3. Why not to directly calculate/apply the radiometric indices over the entire images and then extract the information of interest?. This would be easier and also is the standard in literature.
- Equations 1-7. No need to add the equations in here, you can simply add a table listing the indices and referencing them. These are common in literature.
- General: YOu need to define a block scheme representing your general proposed approach. So far all that I can see is standard application of well-kwnon techniques to your study area. Even the way in which you are applying the different steps is not the standard, making it sound as something rather complex, when these are the very basic steps of every remote sensing process.
RESULTS
- L283. English.
- Table 5. You need to properly explain these results. I do not clearly see the correlation between variables and indices.
- General: While the results are interesting, you are investing a lot of your time and effort on them, with a rather poor to non-existent methodology. There are no clear evaluations of your results and comparisons with the papers that you had cited on the introduction as relevant for this research.
DISCUSSION AND CONCLUSION
- Please separate these two sections. If you must, join results and discussion.
- All the information offered in here could have been better summarized directly on the results, adding a proper analysis of the data in the moment in which it was presenting. At the end you are telling that all the things you have done did not work and that even though you had thought it was possible to use the data for this application, it turned out not to be on that way.
Reviewer 2 Report
This paper examined the relationships between vegetation indices and silk production using PS images, which is very interesting but requires some modifications to the statistical modeling for the consideration of later publication.
Line 351. Multiple regression seems applicable. The authors investigated the relationships between a single index and silk production. However, the estimation or prediction of silk production can be more developed using a (linear or non-linear) regression model. Multiple variables can be used for the regression model if only the tests for multicollinearity of the explanatory variables are passed. Please try the multicollinearity test first, and you can discuss whether simple or multiple regression is more appropriate.
Reviewer 3 Report
The manuscript is devoted to the important issue of studying the possibilities of using satellite data for agricultural monitoring. In particular, the authors consider the possibilities of silkworm monitoring and indirect determination of silkworm productivity.
The article is written in a good language and is logically constructed. It contains new scientific data and can be recommended for publication. The following can be noted as a minor comment:
1. It is necessary to describe the object of research more fully. It is necessary to give data on the size of trees and the distance between them in plantations. Without this it is impossible to evaluate the applicability of the satellite data used.
2. The assumption that the results can be improved by using more detailed images from a drone also looks unfounded. With more detailed spatial resolution, there may be many other factors that affect the image, and the results may be even worse.